# Cellular and Molecular Bases for the Application of Polyphenols in the Prevention and Treatment of Cardiovascular Disease

**DOI:** 10.3390/diseases13070221

**Published:** 2025-07-15

**Authors:** Carlo Caiati, Emilio Jirillo

**Affiliations:** 1Unit of Cardiovascular Diseases, Department of Interdisciplinary Medicine, University of Bari “Aldo Moro”, 70124 Bari, Italy; 2Interdisciplinary Department of Medicine, Section of Microbiology and Virology, School of Medicine, University of Bari “Aldo Moro”, 70124 Bari, Italy; emilio.jirillo@uniba.it

**Keywords:** cytokines, dendritic cells, immunotherapy, macrophages, myocardial infarction, T lymphocytes

## Abstract

Background: Cardiovascular disease (CVD) is very widespread in countries with a Western-style diet, representing one of the major causes of morbidity. Genetic factors, obesity, diabetes, dyslipidemia, smoking, and ageing are risk factors for CVD outcomes. From a pathogenic point of view, the condition of low-grade inflammation of the arteries leads to endothelial damage and atherosclerosis development. Nowadays, a broad range of drugs is available to treat CVD, but many of them are associated with side effects. Therefore, alternative therapeutic remedies need to be discovered in combination with conventional drugs. A balanced diet rich in fruits and vegetables, e.g., the Mediterranean diet, has been shown to lower the incidence of CVD. Plant-derived polyphenols are ingested in food, and these compounds can exert beneficial effects on human health, such as antioxidant and anti-inflammatory activities. Objective: In the present review, the cellular and molecular bases of the beneficial effects of polyphenols in the prevention and treatment of CVD will be pointed out. Methods: This review has been conducted on the basis of a literature review spanning mainly the last two decades. Results: We found that an increased dietary intake of polyphenols is associated with a parallel decrease in chronic disease incidence, including CVD. Conclusion: Despite a plethora of preclinical studies, more clinical trials are needed for a more appropriate treatment of CVD with polyphenols.

## 1. Introduction

Cardiovascular disease represents one of the major causes of morbidity in countries adopting Western lifestyles, with an annual expectation of deaths by 2030 that exceeds 23.6 million [1]. The term CVD encompasses a variety of conditions, such as coronary artery disease (CAD), stroke, peripheral artery disease, hypertension, cerebrovascular disease, and heart failure (HF). Among risk factors of CVD, genetic factors, obesity, diabetes, dyslipidemia, smoking, and ageing account for occurrences of CVD [2,3]. The above conditions lead to endothelial cell dysfunction, oxidative stress, proliferation of smooth muscle cells, and fibroblasts, with conversion of macrophages to foam cells within the artery walls [4]. Furthermore, the condition of vascular low-grade inflammation promotes atherosclerotic plaque formation, ultimately, causing HF [5]. Therapeutically, a broad range of drugs is available for the treatment of CVD, i.e., statins, angiotensin-converting enzyme inhibitors, angiotensin receptor blockers, calcium channel blockers, fibrates, and beta-blockers; however, many of them are associated with side effects [6]. Therefore, there is a need to discover and apply innovative therapies in combination with conventional ones for a more appropriate management of CVD [7,8]. It is known that a balanced diet is beneficial for preventing CVD. In fact, consumption of fruits and vegetables has been shown to decrease the incidence of CVD. In this respect, the Mediterranean diet (MD) decreases inflammatory biomarkers, e.g., interleukin (IL)-1 beta, IL-5, and C-Reactive Protein (CRP), thus preventing chronic disease outcomes [7,9,10,11]. In this framework, the PREDIMED study demonstrated that the MD, based on the high consumption of fruits, vegetables, whole grains, and extra virgin olive oil (EVOO), was associated with a reduced risk of CVD [12]. Of note, dietary interventions aimed at reducing low-grade inflammation have led to divergent results due to differences in the tested dietary compounds and chosen inflammatory markers [13,14]. Plant-derived compounds contained in food possess beneficial effects on human health. Among these natural products, polyphenols can be found in fruits, vegetables, seeds, nuts, as well as in red wine, tea, coffee, extra virgin EVOO, and chocolate [15,16,17,18,19]. Nowadays, the human population is more aware of the beneficial effects of polyphenols, and their dietary intake has increased, with a parallel decrease in chronic disease, including CVD [20]. In the present review, the classification, pharmacological activities, and main mechanisms of action of polyphenols will be described. Experimental and clinical evidence of the beneficial effects of these natural compounds on CVD will be discussed.

### 1.1. Classification and General Properties of Polyphenols

Polyphenols are classified according to the number of phenolic rings and the structural elements they bind [16]. They can be divided into four main classes: flavonoids, non-flavonoid stilbenes, phenolic acids, and lignans (Figure 1) [21]. Flavonoids are naturally occurring compounds that encompass six categories: flavanones, flavones, flavanols, isoflavones, flavan-3-ols, and anthocyanidins (Figure 2) [22]. Structurally, they possess two aromatic rings and a heterocyclic ring with a C6-C3-C6 configuration (Figure 2). They are contained in plants as glycosides and non-glycosylated conjugate compounds, and their type of structure influences bioavailability [23]. Stilbenes, e.g., resveratrol (RES), are composed of two phenyl residues linked by a two-carbon methylene bridge, which can be glycosylated, methylated, or prenylated by specific enzymes (Figure 1) [24]. Among flavonoids, flavonols and flavan-3-ols have been the object of intensive research. The flavonol quercetin exhibits antihypertensive effects by acting on the contraction of smooth muscles in renal blood vessels, producing vasodilation [25]. Among flavan-3-ols, epigallocatechin-3-gallate (EGCG) is present in green tea and is endowed with antioxidant, anti-inflammatory, and antiatherogenic properties [26]. Among stilbenes, RES is the most studied compound for its anti-inflammatory, antioxidant, anti-proliferative, anti-apoptotic, and mitochondrial protective effects [27].

### 1.2. Absorption of Polyphenols

In the stomach, polyphenols are digested by pepsin, and peristaltic movements in the presence of a low pH into particles, even less than 500 microns in diameter [28]. The passage of polyphenols from the stomach to the small intestine occurs at a pH around 7, and then, pancreatic and biliary enzymes become activated [29]. From a general point of view, in food, most phenolic compounds are present in glycosylated form (where the sugar group is called “glycone” and the non-sugar polyphenolic component “aglycone”) or in the form of polymers. Both forms are unable to be absorbed (with some exceptions, such as anthocyanins) and must undergo several processes, especially hydrolysis by the microflora of the colon or by some enzymes before being absorbed; the degree of absorption often derives from the chemical structure of these compounds and from the sugar part linked to them [21]. From their contact with the oral cavity, especially with saliva, many phenolic compounds undergo transformations until the formation of glycosides [30]. There are three further possible mechanisms through which glycosides undergo the hydrolyzation process: (i) in the microvilli present at the level of the epithelial cell membranes of the small intestine by the enzyme lactase phloridizin hydrolase (LPH) [29]; (ii) through the β-glucosidase present in the cytoplasm of epithelial cells where the compounds are transported by the active sodium-dependent glucose transporter (SGLT1) [31]; (iii) the polyphenols that are not absorbed in the small intestine; once they reach the colon, they are hydrolyzed by the microflora present in the colon [32]. Thus, after the hydrolyzation process, polyphenols in the form of aglycons enter the intestinal epithelium according to different modalities. For instance, polyphenols with low molecular weight, i.e., phenolic acids, flavonoid aglycon, tea polyphenols, and cocoa polyphenols (epicatechin, procyanidin B2, and catechin) are absorbed by passive diffusion [33]. Another method of polyphenol absorption is sodium/glucose transport through the sodium/glucose-linked transporter 1 (SGLT1) [34]. Accordingly, glycosides may be absorbed by SGTL1 in small amounts, and then re-secreted into the digestive system, or they may be further digested by a cytosolic glucosidase [35]. Thus, polyphenols can undergo transepithelial transport through a monocarboxylic acid transporter, as in the case of caffeic acid and ferulic acid (Figure 1) [36]. Most polyphenols reach the large intestine, where they undergo significant transformations due to intestinal microbiota. In particular, they are digested by bacteria of the gut microbiota via glycosylation, hydroxylation, demethylation, deconjugation, ring cleavage, hydrolysis, epimerization, and chain shortening processes [37]. Therefore, gut microbiota play a crucial role in their metabolism. These microbial transformations generate bioactive metabolites that can influence the gut environment, modulate immune responses, and potentially contribute to disease prevention [38]. In particular, poorly absorbed wine and tea polyphenols given orally to rats were proven to reduce DNA oxidative damage in caecal mucosal cells [39] and reduce the number of tumors in rats treated with azoxymethane [40].

On the other hand, some polyphenols are absorbed into the enterocytes of the small intestine, and before entering circulation, undergo phase II of enzymatic detoxification with the production of sulfates, glucuronides, and methylated derivatives [41]. Polyphenol bioavailability and accumulation in tissues depend on multidrug resistance-associated proteins, which are ATP-dependent efflux transporters, and referred to as phase III metabolism [42]. Then, polyphenols reach the bloodstream mostly coupled to proteins, and the liver via the portal circulation, where they are conjugated to O-sulphate or O-glucuronide forms (a second phase metabolism), and finally, are eliminated through the kidneys [43] (Table 1).

Other factors involved in polyphenol absorption include the presence of specific dietary components that can significantly impact polyphenol absorption and metabolism. The intestinal absorption of polyphenols can be affected by various factors, including the food matrix and other bioactive compounds. These interactions can either enhance or hinder polyphenol bioavailability, which is the extent to which they become available for absorption and exert their effects. Proteins, lipids (fats), and carbohydrates present in food can interact with polyphenols, potentially impacting their solubility, stability, and ability to be absorbed. For example, some proteins may bind to polyphenols, reducing their availability for absorption, while fats might protect polyphenols from degradation or promote their solubility, especially more hydrophobic ones like curcumin, naringenin, and others [44].

Polyphenols can bind to dietary fiber, affecting their release and subsequent absorption [45].

Methods like boiling, frying, or baking can alter polyphenol content and their interactions with other food components [46].

The gut microbiome plays a crucial role in polyphenol metabolism and absorption. Consuming prebiotics and probiotics can support a healthy gut environment, potentially improving polyphenol bioavailability [47].

In conclusion, a healthy diet can maximize polyphenol absorption, enhancing their positive protective effects.

**Table 1 diseases-13-00221-t001:** Antioxidant Activity of Polyphenols.

1.1. Scavenging activity depends on the donation of an electron or H atom from a hydroxyl group to a free radical [48].

1.2. A catechol group in the structure of polyphenols is associated with antioxidant activity [49].

1.3. The phenolic core of quercetin and catechin scavenges reactive oxygen species (ROS), acting as a buffer or collecting electrons [50].

1.4. Polyphenols inhibit enzymes, such as xanthine oxidase and nicotinamide adenine dinucleotide phosphatase, thus reducing the generation of ROS [51].

1.5. Quercetin exhibits the best capacity to chelate metal ions [52].

### 1.3. Antioxidant Properties of Polyphenols

Polyphenols behave as potent antioxidant agents thanks to catechol groups and hydroxylation patterns, such as the 3-hydroxyl group in flavanols or electron deficiency in anthocyanins [53]. Using the ferric reducing ability power, it has been demonstrated that the presence of a catechol ring in the structure of polyphenols is associated with their antioxidant activity [49]. Reactive oxygen species (ROS), i.e., superoxide, hydrogen peroxide, and hypochlorous acid, are scavenged by quercetin and catechin through the phenolic core, acting as a buffer or collecting electrons [50]. Furthermore, polyphenols have been shown to inhibit enzymes that generate ROS, such as xanthine oxidase and nicotinamide adenine dinucleotide phosphatase [51]. Among polyphenols, quercetin has the best capacity to chelate metal ions due to its low redox potential, thus preventing the production of ROS [27]. The scavenging activity of polyphenols is connected to their ability to donate an electron or H atom from an aromatic hydroxyl group to a free radical, thus abrogating its effect [48]. The antioxidant capacity of polyphenols in vivo is lower than in vitro, since it can be mimicked by other compounds [54]. For instance, in vivo, the polyphenol-mediated antioxidant activity exerted by apple consumption is mostly due to the metabolic effect of fructose on urate.

### 1.4. Effects of Polyphenols on the Vascular Endothelium

The major function of endothelial cells (ECs) is to regulate vascular tone [55]. The endothelial (e) nitric oxide (NO) synthase (eNOS) generates NO from L-arginine, which, in turn, acts on the vascular smooth muscles, thus triggering guanyl cyclase, with accumulation of cyclic guanosine monophosphate, which activates the protein kinase G, thus leading to vasorelaxation. Furthermore, the endothelium-derived hyperpolarizing factor causes vasorelaxation, targeting the K+ channels in the vasculature. Additionally, prostacyclin I2 (PGI2), generated during the cyclooxygenase (COX) pathway, leads to vasodilation. On the other hand, endothelial products, such as angiotensin II (ANG II), endothelin-1 (ET-1), and thromboxane (TXA) A2, have vasoconstrictive effects [56]. NO generation accounts for the main effects of polyphenols on the endothelium [57]. In this respect, red wine polyphenols are a potent inducer of serum NO in healthy subjects 30 min after ingestion [58]. In vitro studies have demonstrated that healthy human peripheral blood monocytes are an additional source of NO, thus contributing to vasodilation after ingestion of red wine [16]. In this regard, short-term oral treatment of normotensive rats with red wine polyphenols decreased blood pressure [59]. Such an effect depends on the induction of the gene responsible for inducible NO synthase and COX-2 in the arteries, as well as on the calcium ion-dependent pathway [60]. In this last respect, RES and quercetin have been shown to induce an increase in calcium concentration by opening the potassium channels or inhibiting Ca^2+^ ATPase within the endoplasmic reticulum of ECs [61]. Evidence has been provided that red wine polyphenols enhance endothelial NO production via the redox-responsive PI3/Akt channel, the increase in intracellular protein-Ca^2+^, and tyrosine phosphorylation with activation of eNOS [62,63]. Apart from NO generation, polyphenols exhibit other effects on the endothelium via increased release of PGI2 [64]. In fact, in vitro and in vivo studies, using cocoa extracts rich in procyanidins, demonstrated that the ratio of leukotrienes-to-PGI2 was reduced. Moreover, polyphenols can increase endothelial NO by decreasing levels of phosphodiesterases (PDE)-2 and PDE-4 [65] (Table 2).

Moving to the clinical arena, strong epidemiologic data support a relationship between higher intake of polyphenolic flavonoids and reduced risk of cardiovascular disease; this seems to be mediated by improved endothelial function and reduced platelet aggregation, as appropriately reviewed [66].

While epidemiological studies and clinical trials can show associations between polyphenol intake and health outcomes, they often lack the detail to explain the underlying mechanisms. Mechanistic studies, especially those using endothelial cell models, can pinpoint the precise molecular pathways and cellular processes that polyphenols affect. We think that the best approach for this purpose is to study the endothelial function in clinical trials by using either the FMD approach, which, however, does not precisely reflect the coronary endothelial function [67], or the direct non-invasive coronary endothelial function test with transthoracic coronary Doppler echocardiography, using the cold pressure test as endothelium-dependent stimulus [68,69,70].

### 1.5. Anti-Inflammatory Activity of Polyphenols

Inflammation is a response of the body to various stimuli, including pathogens, mechanical insults, and damaged tissue. Pro-inflammatory cytokines, e.g., interleukin (IL)-1 beta, IL-6, IL-8, and tumor necrosis factor (TNF)-alpha, as well as various enzymes, such as COX, lipooxigenase (LOX), and protein kinase, are responsible for the inflammatory response. With special reference to the role of polyphenols, there is evidence that red wine polyphenols can, in vitro, reduce the production of pro-inflammatory cytokines, blocking the activation of the NF-kB pathway [71]. Moreover, red wine polyphenols can interfere with endotoxin binding to Toll-Like Receptor (TLR)-4, thus abrogating the nuclear factor kappa-light chain enhancer of activated B cells (NF-kB) pathway with interruption of pro-inflammatory cytokine release [72]. Additionally, polyphenols contained in fermented grape marc (FGM) induce activation of Foxp3+ T regulatory cells with release of the anti-inflammatory cytokine, IL-10 [73]. In addition, FGM reduces the respiratory burst of neutrophils and basophils in in vitro experiments, playing an antioxidant and anti-inflammatory role [74]. Quercetin has been found to dampen the generation of prostaglandins, leukotrienes, and TAXs, abrogating production of COX and LOX [75,76]. In fact, both COX and LOX mediate the formation of arachidonic acid, which, in turn, fuels inflammation via release of IL-1 beta and IL-8. The nucleotide-binding domain and leucine-rich repeat-containing receptors (NLRs) belong to the family of pattern recognition receptors (PRRs), triggering inflammatory responses upon danger and cell damage signals. Among them, NLRP3 inflammasome is a multiprotein complex that activates inflammatory caspase-1 [77]. Caspase-1 cleaves and maturates the pro-inflammatory cytokines IL-1 beta and IL-18, as well as the protein gasodermin, contributing to the release of the above mediators, thus initiating the cell death pyroptosis [78]. Activation of the NLRP3 inflammasome is involved in CVD, including atherosclerosis, myocardial infarction, and cardiac remodeling [79]. In this framework, in the middle cerebral artery occlusion/reperfusion model, supplementation of various polyphenols decreased levels of NLRP3 [80]. This is associated with the downregulation of IL-1 beta and IL-18 in the serum or brain tissue [81]. In the myocardial ischemia (MI)/reperfusion model, certain polyphenols, i.e., RES and flavone, in vivo reduced levels of caspase-1 and IL-1 beta in the myocardial tissue [82,83]. In all these studies, the decrease in NLRP3 levels was associated with improvement of clinical markers [80]. Clinically, aged male subjects at high cardiovascular risk underwent acute administration of aged wine, with a decrease in *Tlr2*, *Nlrp3*, and *Il1receptor* genes [84] (Table 3).

These data have been confirmed in clinical trials. In healthy adults, 4 weeks of red wine intake (with high RES content) significantly reduced PCR and IL-6 levels and improved the response of platelet and leukocyte adhesion molecules (sP-selectin and sE-selectin) [85,86,87]. However, considering resveratrol as a supplement, a meta-analysis of 11 randomized controlled trials observed no effect on PCR concentration [88]. As usual, the supplementation, not being a natural intervention, creates problems and appears less effective. Natural products are always better. In fact, short-term RCTs (1–4 weeks) focusing on blueberries, grapes, apples, and pomegranates showed a significant reduction in inflammation markers, including IL-6, PCR, and tumor necrosis factor (TNF-alfa) in healthy or hyperinsulinemic subjects at high cardiometabolic risk [89,90,91,92,93,94,95].

Further research is needed to optimize polyphenol delivery, determine optimal dosages, and understand individual responses to polyphenol interventions.

### 1.6. Anti-Atherogenic Effects of Polyphenols

Atherosclerosis represents a pathogenic common denominator of various diseases, including CAD, ischemic stroke, and peripheral artery disease [96]. This disease stems from the endothelial damage provoked by several offending factors that then drive the augmentation of ROS in the blood. Those offending factors have been recently reported in [7,97]. In brief, they are largely man-made, such as stress, pollutants of all sorts (especially those contained in food, such as farming chemicals, fertilizers, and pesticides and herbicides like glyphosate), drugs, processed food, tabacco smoking, air pollution, alcohol, cosmetic and cleaning products, heavy metals, chronic infections, electromagnetic radiation (cellular phone, cell-tower emitting radiation), ionizing radiation (in particular those medically derived like computed tomography scan and angiography), and intravascular prosthesis like arterial stents. Diabetes per se induces tissue damage, and it terribly enhances the damaging effects of the previously mentioned atherogenic factors, dramatically enhancing the formation of ROS [7]. From a pathogenic point of view, increased levels of ROS further enhance endothelial damage, with the intervention of neutrophils, macrophages, and platelets [98]. In fact, prolonged contact of ECs with hydrogen peroxide, peroxynitrite, and oxidized low-density lipoproteins (ox-LDL) leads to severe damage of the endothelium [7,96,99]. One of the initial consequences of coronary endothelial dysfunction is the reduction of NO production and the ensuing microvascular vasoconstriction at rest. This kind of derangement can be spotted with positron emission tomography (PET) since it causes myocardial dishomogeneous perfusion at rest [100] and is the mechanism that explains the angiographic slow coronary phenomenon, as recently demonstrated [101].

This strong oxidative drive also involves oxidated LDL microparticles. This causes the first step of atherosclerotic plaque formation, that is, the generation of oxidized LDLs, which pass through the endothelial barrier, eliciting cytotoxic effects and the inflammatory response [102], since ox-LDL microparticles are modified substances that elicit a strong immunologic reaction. Focusing on the molecular biology details, activated ECs express adhesins, i.e., vascular cell adhesion-1 (VCAM-1), intercellular adhesion molecule 1 (ICAM-1), and E-selectins, which allow transmigration of monocytes and T cells into the arterial wall [103]. Particularly, monocytes engulf ox-LDL, becoming foam cells, which accumulate as fatty bands in the artery walls [104]. Then, a stabilized plaque is formed, which can break under prolonged inflammation [105]; this happens only in case of prolonged and uncontrolled exposure to those factors generating ROS with consequent escalating concentration of ox-LDL. Ruptured plaques cause thrombosis, which may lead to heart attacks, ischemic strokes, and peripheral ischemia [106]. There is evidence that polyphenols can exert beneficial effects on atherosclerosis. In cholesterol-fed rabbits, administration of red wine polyphenols decreased neointimal growth, lipid accumulation, and inflammation in the iliac arteries [107]. In hamsters, red wine supplementation reduced neointimal hyperplasia, inhibiting the entry of monocytes into the arterial wall [108]. Clinically, there is evidence that purple grape juice reduced the levels of plasmatic ox-LDL in patients with CAD [109]. This effect has been shown to depend on the production of NO by polyphenols, as also supported by others [110,111] (Table 4). This result has been confirmed in a controlled clinical trial [112]. In this study, the traditional Mediterranean diet with its high content of polyphenols [113] was proved to be beneficial in reducing the extent of LDL oxidation in subjects with a high risk of CAD.

However, the main radical approach that can stop progression and start regression of atherosclerosis is the elimination of those damaging factors (mentioned before) that create ROS and cause chronic endothelium inflammation and ox-LDL [114].

## 2. Focus on the Cardiovascular Effects of Relevant Polyphenols

### Flavan-3-Ols

Flavan-3-ols (Figure 2) represent the most abundant polyphenols in fruits, vegetables, red wine, green tea, and cocoa [115]. They encompass monomeric, oligomeric, and polymeric compounds. Monomeric forms include catechin, epicatechin, gallocatechin, epigallocatechin, epicatechin-3-O-gallate, and EGCG (Figure 2). Oligomers or polymers are known as proanthocyanidins, while polymers composed of epicatechin or catechin are termed procyanidins. The antioxidant activity of flavan-3-ols is based on their ability to donate an electron or to chelate metal ions, thus stopping ROS production [116]. At the same time, flavan-3-ols maintain mitochondrial activity while enhancing antioxidative enzymes involved in ROS scavenging [117,118]. The anti-inflammatory activity of flavan-3-ols depends on the regulation of gene expression involved in cardiometabolic health. Particularly, they act on endothelial transcription factor GATA-2, the NF-kB p105 subunit, forkhead box C1, and peroxisome proliferator-activated receptor gamma [119]. In addition, flan-3-ols target different miRNAs, regulating cellular pathways involved in cell adhesion, cellular signaling, and immune response [120]. Of note, cocoa flavan-3-ols metabolites enhance *ApOAI* expression, which represents the major component of high-density lipoproteins, thus exerting antiatherogenic properties [121]. The cardioprotective effects of flavan-3-ols have been attributed to two major microbial-derived metabolites, namely, the hydroxy-phenyl-gamma-valerolactones, and their derived hydroxy-phenyl valeric acids [122]. These metabolites have, in vivo, shown hypotensive activity in rats, and in vitro abrogation of monocyte adhesion to ECs treated with tumor necrosis factor (TNF)-alpha [123]. Another flavan-3-ol metabolite, the protocatechuic acid, diminished diabetic cardiomyopathy in rats, stimulating glucose metabolism, improving oxidative stress, and reducing inflammation [124]. Flavan-3-ols have been shown to act on gut dysbiosis, improving cardiac function. In fact, a metabolite from the gut microbiota, trimethylamine N-oxide (TMAO), has been associated with CVD pathogenesis in terms of increased cholesterol levels and higher risk of atherosclerosis [125]. In this respect, the intake of cocoa and red berry flavan-3-ols reduces TMAO levels, improving cardiovascular markers in healthy aging adults [126]. Various clinical trials have been conducted, using cocoa flavan-3-ols in patients with CVD. In hypertensive individuals, consumption of dark chocolate led to a reduction of systolic blood pressure (SBP) and diastolic blood pressure (DBP) in comparison to baseline [127]. In another study, in essential hypertensive subjects with impaired glucose tolerance receiving 100g/day of chocolate, a reduction in both SBP and DBP and an increase in flow-mediated dilation (FMD) were observed [128]. In patients with CAD receiving dietary high-flavan-3-ol intervention, an increase in brachial artery FMD and a reduction in SBP were recorded [129]. In patients with congestive HF, intake of flavan-3-ol-rich chocolate improved FMD [130]. Other studies have been conducted with green tea catechins in both healthy and unhealthy individuals. In healthy volunteers aged between 21 and 70 years receiving two capsules of *Camelia sinensis* for 3 months, a reduction in SBP was observed [131]. In another group of healthy adult men aged 18–35 years, administration of 450 mg of sour tea led to a reduction in SBP and DPB [132]. In healthy postmenopausal women, acute ingestion of catechin-rich green tea improved postprandial glucose status while increasing serum thioredoxin levels, but no changes in cardiovascular risk factors were observed [133]. In overweight women aged 19–57 years, receiving a low-calorie diet along with 3 capsules of green tea or placebo capsules, a decrease in SBP and DPB was observed in both groups [134]. Another trial conducted in healthy male volunteers supplemented with an aqueous green tea extract showed no alterations of cardiac risk factors [135]. Additionally, minor effects on cardiovascular risk markers were observed following tea catechin administration to active older people [136]. Taken together, the above data suggest that studies with polyphenols conducted in both healthy and unhealthy individuals led to contrasting results.

## 3. Resveratrol

Stilbenes, and particularly, RES (Figure 1), despite a low bioavailability, possess a strong antioxidant activity in vitro. RES protects cardiomyocytes and ECS against ROS effects, inhibiting NADPH oxidases while increasing the mitochondrial respiratory chain enzymes [137]. RES acts by upregulating SIRT1, which, in turn, induces deacetylation of NF-kB, and enhancement of superoxide dismutase (SOD), catalase, and glutathione peroxidase 1 [138,139]. Furthermore, RES can reverse eNOS uncoupling, upregulating GCH1 expression in apolipoprotein E knockout mice [140]. Additionally, RES activates Nrf-2, which, in turn, increases cellular antioxidant content in the placenta of sows and piglets [141]. As a potent anti-inflammatory agent, RES can inhibit the expression of pro-inflammatory cytokines, downregulating TLR4 expression and silencing NF-kB activity [142,143]. Moreover, RES can inhibit VCAM-1, ICAM-1, and E-selectin, suppressing the TNF-alpha-induced NF-kB activation [144]. RES inhibits COX-1 and COX-2 enzymes via SIRT1 activation, thus decreasing PGE2 and TAX2, and, consequently, inflammation [145]. In patients with systolic HF, RES administration improved clinical conditions by inhibiting oxidative phosphorylation in leukocytes, gene expression encoding B cell receptors, and leukocyte extravasation signal [52]. ROS-mediated overexpression of MAPKs is involved in cardiac hypertrophy and remodeling [146]. RES can stimulate MKP-1 and downregulate mTOR, thus dampening mitogen-activated protein kinase (MAPK) activity, with a reduction in cardiac and endothelial hypertrophy [147,148]. With special reference to cardiac fibrosis, it has been reported that RES can mitigate cardiac fibroblast activity in rats, downregulating the transforming-growth factor-beta/Smad 2/3 signaling pathway via overexpression of the Smad 7 inhibitor protein and silencing the miR-17 gene [149]. RES can modulate endothelial function, inhibiting overproduction of the vasoconstrictive agent ET-1 and enhancing eNOS phosphorylation, with an increase in NO production [150]. There is evidence that upregulation of ET- and a decrease in NO are involved in the pathogenesis of atherosclerosis [151]. The effects of RES on mitochondrial biogenesis have been documented. In fact, RES activates the AMPK/SIRT1/PGC1 alpha pathway, with enhancement of Nrf-1 and Nrf-2 transcription factors, thus attenuating high-glucose oxidative stress and cardiomyocyte apoptosis in diabetic mice [152]. As far as clinical trials are concerned, the effects of RES have been studied in patients with hypertension. In one study, long-term administration of RES could reduce hypertension along with standard medical therapy [153]. In a meta-analysis, in hypertensive subjects, daily RES consumption reduced SBP, but not DBP [154]. Conversely, in two other studies, the hypotensive effects of RES were not confirmed [52,155]. With special reference to vascular protection, RES long-term administration improved the FMD of the brachial artery in overweight and hypertensive individuals, stable CAD patients, and patients with metabolic syndrome, respectively [156,157,158]. In another study, acute RES administration to hypertensive patients improved FMD without changes in SBP [159]. A systematic review and meta-analysis have provided evidence that RES can modify lipid profile, diabetes, and inflammation associated with atherosclerosis in metabolic syndrome patients [160,161,162]. A few clinical trials have been conducted in patients with HF. In post-MI patients, administration of 10 mg/day of RES for 3 months improved the diastolic function [157]. In patients with angina pectoris, RES supplementation at 20 mg/day for 2 months reduced serum levels of the N-terminal prohormone brain natriuretic peptide (NT-proBNP) [163]. In patients with symptomatic systolic HF, 100 mg/day of RES supplementation improved systolic and diastolic function, as well as serum biomarkers such as NT-proBNP and IL-1 and Il-6 levels [52].

## 4. Curcumin

Curcumin (diferuloyl methane) is a natural polyphenol extracted from the rhizomes of the turmeric plant *(Curcuma longa* L.) [164]. Structurally, curcumin possesses a constitutional double bond, thus behaving as an electron donor, which mitigates ROS effects [165]. Furthermore, curcumin exerts anti-inflammatory effects, as well as modulation of lipid metabolism and the immune system [166]. In cadmium-induced hypertensive rats, curcumin administration normalized vascular dysfunction and blood pressure [167]. Similar results were achieved in Sprague rats with lead acetate- and cadmium chlorate-induced hypertension [168]. Furthermore, in spontaneous hypertensive rats, curcumin administration attenuated the coronary artery damage [169]. Additionally, in an ANG-II-induced hypertensive rat model, curcumin administration reduced the ANG-II type-I receptor-mediated vasoconstriction, thus preventing hypertension [170]. A few clinical trials have been conducted in hypertensive patients using curcumin. A group of 14 men and 24 women with an average blood pressure of 121–140/81–90 mm Hg received cavacurmin (500 mg), eicosapentaenoil acid, astaxanthin, and gamma linolenic acid for 4 weeks [171]. A significant decrease in SBP was observed only in women. In refractory or relapsing lupus nephritis patients, administration of curcumin (500 mg) for 3 months led to a significant decrease in SBP [172]. Moreover, a combination of curcumin and galactomannan (500 mg) was administered to obese subjects, with a declining trend in blood pressure and aortic stiffness and an increase in anti-inflammatory cytokines [173]. Conversely, in another study, a 12-week treatment of healthy middle-aged and older adults with 200 mg of curcumin did not modify blood pressure, despite a reduction in oxidative stress and improvement in endothelial function [174]. Previously, evidence has been provided that RES and curcumin in combination could lower oxidative stress, inflammation, and tumor growth [175]. Such a combination improved endothelial function, inhibiting the gene regulatory activity of TNF-alpha and abrogating the NF-kB pathway.

## 5. Extra Virgin Olive Oil Polyphenols

EVOO represents a food supply endowed with antioxidant and anti-inflammatory activities [176]. EVOO is mainly composed of monosaturated fatty acids, alpha-tocopherol, and polyphenols [177]. The phenylethanoid derivatives hydroxytyrosol (HT) and thyrosol are the major polyphenols contained in EVOO [178]. HT is the most studied EVOO polyphenol in terms of anti-inflammatory activity and CVD prevention. In healthy male Wistar rats, HT administration inhibited collagen-induced platelet aggregation in whole blood [179]. This effect has been attributed to the inhibition of platelet synthesis of TxB2, production of vascular PGI2, and an increase in vascular NO. Additionally, HT alkyl ether derivatives exerted similar effects, thus acting as anti-aggregating agents at the endothelial level [180]. In human clinical trials, HT has been studied for its capacity to attenuate the pathogenesis of atherosclerosis. In 30 hypercholesterolemic volunteers (aged 20–70 years), administration of HT derived from Coratina olives led to a normalization of SBP and lipid profile [181]. Similar results were achieved through supplementation of Body Lipid, containing HT, berberine, coenzyme Q10, and monacolin K to hypercholesterolemic individuals [182]. In another study, administration of HT and punicalagin to an adult population improved dyslipidemia and decreased SBP and DBP [183]. HT and punicalagin increased endothelial capacity and reduced ox-LDL. Furthermore, in 40 healthy volunteers, administration of HT (15 mg/day for 3 weeks) increased in blood samples antioxidant activity, oxidation biomarkers (thiols), and SOD1, while malonedialdehyde (MDA) and NO metabolites were decreased [184]. Conversely, in another study, administration of HT to human volunteers with mild hyperlipidemia did not influence CVD biomarkers, while levels of vitamin C increased [185]. In this framework, a very recent study based on the supplementation of 15 mg HT/day to patients 24 h after stroke for 45 consecutive days led to encouraging results [186]. In fact, a decrease in glycated hemoglobin and DBP and a modulation of the expression of the gene encoding for apolipoproteins were recorded. A limitation of these studies is the possible co-presence of other compounds that can also contribute to the efficacy of the treatment. This is the case of trials conducted with dietary supplementation of EVOO, where the effects of polyphenols cannot be distinguished from those of other components, such as unsaturated fatty acids.

## 6. Cardiovascular Effects of Wheat Polyphenols

Wheat (*Triticum* sp.) is widely used all over the world. There is evidence that 2–3 servings/day of whole wheat grains reduce the risk of CVD [187]. Among phenolic acids, ferulic acid is the major component of wheat, and the number of hydroxyl groups correlates with its antioxidant potential [188]. Experimentally, extracts enriched in ferulic, synaptic, and p-coumaric acids downregulated pro-inflammatory cytokines and chemokine/interferon-gamma-inducible protein 10 kDa [189]. Furthermore, fermented wheat germ polyphenols could reduce lipid metabolism in hyperlipidemic rats, activating the AMPK pathway. Clinically, wheat aleurone improved redox status in overweight/obese individuals at higher risk of CVD [190]. Ferulic acid could lower total cholesterol, triglycerides, MAD, and C-RP in hyperlipidemic individuals, thus preventing atherosclerosis outcomes [191]. The role of quercetin, a flavonol contained in whole wheat grain, has been preclinically investigated. Its athero-protective effects have been ascribed to the suppression of inflammation and apoptosis [192]. Quercetin derivatives can induce regression of atheromatous plaques, triggering autophagy and inhibiting the breakdown of elastin, macrophage infiltration, and production of both matrix-metallo-proteinase 9 and adhesion molecules [193,194]. Additionally, quercetin could prevent cardiac/ischemia and/or reperfusion injury through regulation of the PI3K/Akt pathway [195] (Table 5).

### 6.1. Adverse Effects of Polyphenols

A few side effects attributed to polyphenol administration have been recorded. For instance, RES administration to humans may lead to emesis, mild hepatic dysfunction, and diarrhea [196,197]. In rats, high oral doses of RES (3g/Kg/day) provoked nephrotoxicity [198]. Additionally, flavonoids can cause mild gastrointestinal symptoms, insomnia, headache, palpitations, and an increase in serum transaminases [199,200]. Other side effects of polyphenol ingestion include a reduced gastrointestinal transport of folic acid, thiamine, and iron [201].

There are other toxicity considerations. Polyphenols’ long-term safety data are still being investigated, particularly for high doses. Some polyphenols can exhibit carcinogenic or genotoxic effects at high concentrations, and interactions with medications or other substances are also possible. In particular, high doses of certain polyphenols like caffeic acid have been linked to tumor development in animal studies [202]. Polyphenols can interfere with iron absorption, potentially leading to iron deficiency in susceptible individuals. They can also interact with medications, potentially affecting their efficacy or increasing side effects [203]. Some polyphenols can act as pro-oxidants, potentially contributing to cellular damage under certain conditions through the formation of quinone, which can react with cysteine in glutathione or proteins. This process generates ROS and can cause DNA damage [204]. High intake of certain polyphenols, particularly from supplements, has been associated with liver toxicity [205]. Important factors to consider include the following:-The amount of polyphenols consumed is crucial. Natural dietary sources generally provide safe levels, but high-dose supplementation raises concerns [206].-Metabolic differences between individuals can influence how polyphenols are processed and their effects [207].-More long-term human studies are needed to fully assess the safety of various polyphenol intakes.-Polyphenols in fruits, vegetables, and other plant-based foods are generally considered safe due to lower concentrations and the presence of other beneficial compounds [44].-High-dose polyphenol supplements require careful consideration due to the potential for adverse effects [205].

In summary, while polyphenols offer potential health benefits, especially from dietary sources, it is important to be aware of the potential risks associated with high-dose supplementation and to consider individual factors when assessing polyphenol intake.

### 6.2. Caveats in Polyphenols

In clinical trials, polyphenols, in particular RES, a compound found in red wine and other sources, have shown promising results in preclinical studies for various health benefits, but clinical trials often yield inconsistent results [208]. This discrepancy can be attributed to factors like varying dosages, individual differences in absorption and metabolism, the complexity of translating findings from animal models to humans, and, finally, associated food in the diet; unhealthy food, in fact, can blunt the positive effect of polyphenols. In particular, diets high in processed foods, saturated fats, sugars, and excessive alcohol can negatively impact the gut microbiome, which is crucial for polyphenol absorption and utilization [209]; these variables could be relatively difficult to control in clinical trials.

In the context of polyphenol consumption patterns, the beneficial effects of red wine, fruit juices, green tea, and chocolate have been reported previously. However, it should be stressed that the health impact of such kinds of food depends not only on polyphenol content, but also on consumption pattern, dose, and dietary context. For example, the culture of drinking red wine with meals, as part of the Mediterranean diet, differs significantly from unhealthy patterns of alcohol consumption [210]. Similarly, frequent intake of fruit juices or sweetened tea may contribute to metabolic disorders due to high fructose content [211]. Therefore, it is essential to emphasize that quantity, context, and overall healthy lifestyle are key factors modulating the effects of polyphenols.

Regarding how polyphenols interact with medications, polyphenols, found in many plant-based foods and supplements, can interact with certain medications, potentially affecting how the body processes and utilizes those drugs. These interactions can alter drug metabolism, absorption, and excretion, possibly leading to reduced drug efficacy or increased toxicity. This can be particularly critical for drugs like direct oral anticoagulants (DOACs) with a narrow therapeutic index; supplements containing polyphenols alongside DOAC therapy carry the risk of bleeding or a reduction in DOAC therapeutic effect. In fact, they have antiplatelet effects, which in conjunction with DOACs can potentially significantly increase the risk of bleeding, as is obviously the case when combining antiplatelet drugs with DOACs [212]. These interactions primarily occur through the modulation of drug metabolism and absorption by affecting enzymes like CYP450 and transporters like P-glycoprotein. Some polyphenols can inhibit or induce these enzymes and transporters, leading to altered drug levels in the body. This is particularly critical for those drugs with a narrow therapeutic index, like the above-mentioned DOAC, but also warfarin, cyclosporine A, and digoxin [213]. However, concerns regarding medication interactions arise only with the promotion of polyphenol consumption at levels far above natural occurrence, that is, when they are assumed as supplements [214].

## 7. Conclusions and Future Trends

There is a large body of evidence that polyphenols exert antioxidant and anti-inflammatory activities, thus regulating major pathways involved in cellular activation and metabolism. In this respect, polyphenols exert beneficial effects on CVD, such as stroke, hypertension, and HF. For example, the MD is a balanced diet, which promotes human health, including prevention of CVD. However, dietary foods contain many compounds, e.g., vitamins, minerals, polyphenols, and unsaturated fatty acids, all endowed with protective effects in the host. Therefore, in this review, emphasis has been placed on the cardioprotective effects of single polyphenols alone or a combination, to rule out potential effects of other dietary compounds. Undoubtedly, preclinical studies conducted with a variety of polyphenols suggest their beneficial effects on CVD. On the other hand, clinical trials are still few and, sometimes, based on a low number of participants. Therefore, the actual effects of polyphenol intake on healthy and unhealthy human populations need a more robust confirmation with more clinical trials.

## Figures and Tables

**Figure 1 diseases-13-00221-f001:**
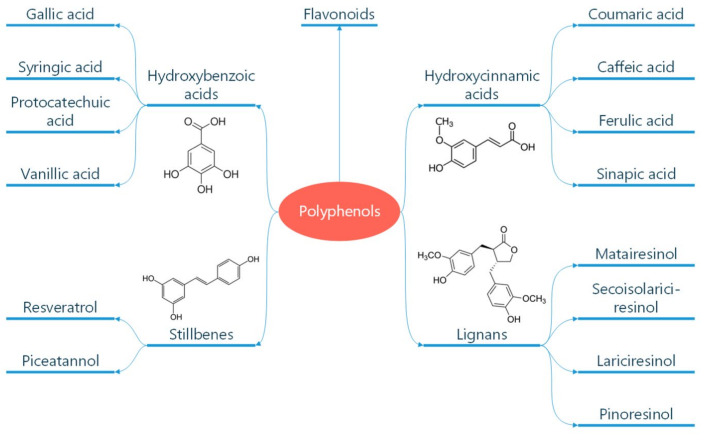
Classification of polyphenols. Polyphenols are natural compounds found in plant-based foods and beverages. Their classification into different subclasses like phenolic acids, flavonoids, stilbenes, and lignans is reported. The chemical formula of these molecules is also reported. Reproduced with permission from Caiati et al. [7].

**Figure 2 diseases-13-00221-f002:**
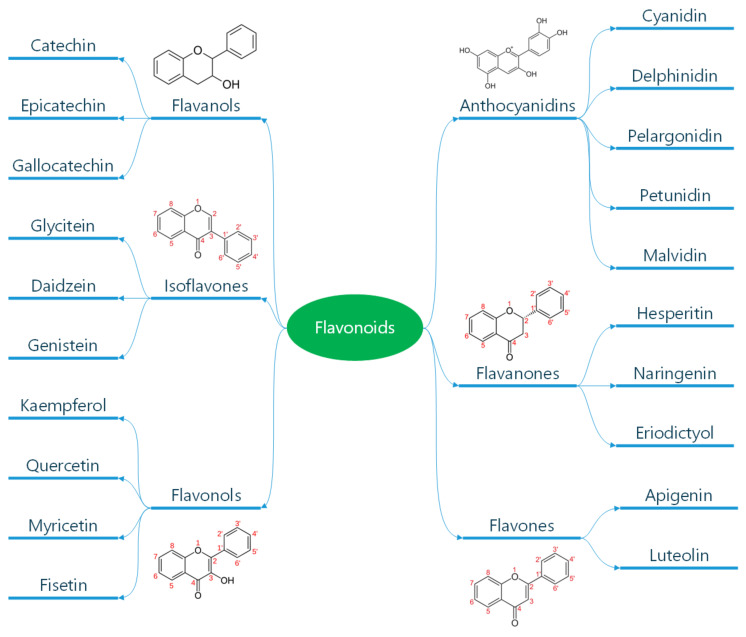
Classification of flavonoids. Flavonoids are a subclass of polyphenols and can be classified into flavonols, flavones, isoflavones, flavanones, antho-cyanidins, and flavanols based on their ring structure as illustrated here. Flavonoids have diverse biological activities and potential health benefits, including antioxidant and anti-inflammatory effects. Reproduced with permission from Caiati et al. [7].

**Table 2 diseases-13-00221-t002:** Effects of Polyphenols on the Vascular Endothelium.

2.1. Polyphenol-induced nitric oxide (NO) generation from endothelial cells and monocytes contributes to artery vasodilation [16,57,58].

2.2. In rats, ingestion of red wine polyphenols generates hypotension through activation of inducible NO synthase, cyclooxygenase-2, and calcium ion-dependent pathway in the arteries [60,61].

2.3. Red wine polyphenols trigger endothelial NO production via the PI3/Akt pathway, the increase in intracellular protein-Ca^2+^, and tyrosine phosphorylation [62,63].

2.4. Cocoa extracts rich in procyanidins cause vasodilation via increased release of prostacyclin I2 [64].

2.5. Polyphenols increase endothelial NO by decreasing phosphodiesterase (PDE)-2 and PDE-4 [65].


**Table 3 diseases-13-00221-t003:** Anti-Inflammatory Activity of Polyphenols.

3.1. Red wine polyphenols reduce the production of pro-inflammatory cytokines, inhibiting the NF-kB pathway, and/or activating T regulatory cells, with release of the anti-inflammatory cytokine interleukin (IL)-10 [16,73].

3.2. Fermented grape marc reduces the respiratory burst of human neutrophils and basophils [74].

3.3. Quercetin decreases the release of IL-1 beta and IL-8, abrogating the generation of cyclooxygenase and lipoxygenase [75,76].

3.4. Polyphenols dampen the activity of the inflammasome NLRP3, with downregulation of caspase1, IL-1 beta, and IL18 [79,80,81,82].

3.5. Reduction of NLRP3 is associated with improvement of clinical markers, as seen in aged male subjects at high cardiovascular risk following acute administration of red wine [80,84].

**Table 4 diseases-13-00221-t004:** Anti-Atherogenic Effects of Polyphenols.

4.1. In cholesterol-fed rabbits and in hamsters, administration of red wine polyphenols decreases neo-intimal growth, lipid accumulation, and entry of monocytes in the iliac arteries [107,108].

4.2. In patients with coronary artery disease, supplementation of purple grape juice reduces levels of oxidized lipoproteins through the generation of nitric oxide [109,110,111].

**Table 5 diseases-13-00221-t005:** Cardiovascular Effects of Polyphenols.

** *5.1. Flavan-3-Ols* **
5.1a Flavan-3-ols metabolites, hydroxy-phenyl-gamma-valerolactones, hydroxy-phenyl valeric acid, and protocatechuic acid exhibit hypotensive activity in rats and decrease diabetic cardiomyopathy, with a reduction in inflammatory biomarkers [122,123,124].

5.1b Cocoa flavan-3-ols supplementation reduces trimethylamine-N oxide in healthy individuals, systolic blood pressure (SBP) and diastolic blood pressure (DBP) in hypertensive individuals, and in patients with coronary artery disease while increasing flow-mediated dilation (FMD) [127,128,130].

5.1c Administration of green tea catechins to healthy volunteers decreased SBP and DBP and improved postprandial glucose status while lowering serum thioredoxin levels [131,132,133].

5.1d No effects of green tea catechin supplementation were observed in healthy male volunteers, active older people, and overweight women [134,135,136].

** *5.2. Resveratrol (RES)* **
5.2a In rodents, RES mitigates cardiac, endothelial hypertrophy, and cardiac fibrosis, dampening MAPK activity and transforming-growth factor-beta/Smad 2/3 signaling pathway [147,148,149].

5.2b RES inhibits endothelin-1, with production of nitric oxide and prevention of atherosclerosis [151].

5.2c In diabetic mice, RES attenuated high-glucose oxidative stress and cardiomyocyte apoptosis through enhancement of Nrf-1 and Nrf-2 transcription factors [152].

5.2d In patients with hypertension, RES administration reduced hypertension [153,154], while in two other studies, such an effect was not confirmed [52,155,170].

5.2e In hypertensive patients, stable coronary artery disease patients, and patients with metabolic syndrome, long-term RES administration improved the FMD of the brachial artery [156,157,158,159].

5.2f RES administration can modify the lipid profile, diabetes, and inflammation in patients with atherosclerosis [160,161,162].

5.2g In patients with heart failure, RES administration improved both systolic and diastolic function, reducing the serum levels of the N-terminal prohormone brain natriuretic peptide [52,157,163].

** *5.3. Curcumin* **
5.3a In hypertensive rat models, curcumin administration normalized vascular function, attenuating coronary artery damage [167,168,169,170].

5.3b In hypertensive patients, refractory or relapsing lupus nephritis patients, and obese subjects, curcumin reduced blood pressure, with an increase in anti-inflammatory cytokines [171,172,173].

5.3c In another study, curcumin did not modify blood pressure in healthy middle-aged and older adults [174].

** *5.4. Extra Virgin Olive Oil (EVOO)* **
5.4a Hydroxytyrosol (HT) inhibited platelet aggregation in rats, decreasing thromboxane B2 and prostacyclin, while increasing nitric oxide [179,180].

5.4b In hypercholesterolemic individuals, HT administration normalized the lipid profile, with a reduction in SBP and DBP [181,182,183]. In another administration, HT did not modify lipid profile and cardiovascular biomarkers [185].

5.4c In patients with stroke, administration of HT 24 h after stroke decreased glycated hemoglobin and DPB [186].

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
