# Peer review of "Cellular and Molecular Bases for the Application of Polyphenols in the Prevention and Treatment of Cardiovascular Disease"

_diseases, 2025, doi:10.3390/diseases13070221_

Round 1
Reviewer 1 Report
Comments and Suggestions for Authors
Dear Author,
The writers of this manuscript have presented a comprehensive overview of the subject in addressed. The authors have managed to incorporate a wide range of relevant studies, which greatly strengthens the credibility of the findings presented. Overall, the paper is well-structured, and the information is quite helpful, making it a useful resource for researchers and healthcare practitioners. Its depth, clarity, and forward-looking approach make it a balanced perspective.
It would be beneficial to make some more adjustments to the language.
With my compliments and best regards,
Author Response
Reviewer 1
The writers of this manuscript have presented a comprehensive overview of the subject in addressed. The authors have managed to incorporate a wide range of relevant studies, which greatly strengthens the credibility of the findings presented. Overall, the paper is well-structured, and the information is quite helpful, making it a useful resource for researchers and healthcare practitioners. Its depth, clarity, and forward-looking approach make it a balanced perspective.
It would be beneficial to make some more adjustments to the language.
With my compliments and best regards,
Answer: many thanks for your appreciation
Reviewer 2 Report
Comments and Suggestions for Authors
This manuscript presents a very interesting and valuable overview of the potential cardioprotective properties of polyphenols. The topic is timely and relevant, both from the perspective of basic research and possible clinical implications.
The authors clearly present the pathophysiological background of cardiovascular diseases, emphasizing the role of chronic inflammation and oxidative stress in the development of atherosclerosis. The classification of polyphenols and the mechanistic discussion are well-structured and comprehensive. Importantly, the manuscript is well-organized, the figures are informative, and the discussion is coherent.
One of the strengths of this review is the use of up-to-date literature. Over 60% of the cited references originate from the last five years, which reflects the current state of knowledge in the field.
Nevertheless, I would like to raise a few points that, while minor, should be addressed to improve the scientific robustness of the manuscript:
1. A large portion of the data presented in the review is derived from preclinical studies using animal models. While this is understandable, it should be explicitly stated that the concentrations of compounds such as resveratrol used in these models often significantly exceed those achievable in humans. Consequently, the translational potential of these findings is limited. This issue has been increasingly emphasized in recent literature, which often describes the gap between promising preclinical results and disappointing clinical outcomes.
2. In several sections, the manuscript highlights the beneficial effects of red wine, fruit juices, green tea, and chocolate. However, it should be stressed that their health impact depends not only on polyphenol content, but also on consumption pattern, dose, and dietary context. For example, the culture of drinking red wine with meals, as part of the Mediterranean diet, differs significantly from unhealthy patterns of alcohol consumption. Similarly, frequent intake of fruit juices or sweetened tea may contribute to metabolic disorders due to high fructose content. Therefore, it is essential to emphasize that quantity, context, and overall healthy lifestyle are key factors modulating the effects of polyphenols.
3. Another important but unaddressed issue is the possibility of interactions between polyphenols and commonly used medications. For example, recent studies have shown that certain polyphenols may interfere with the pharmacokinetics of direct oral anticoagulants (DOACs), potentially altering their efficacy or safety. This should be briefly mentioned, as it is particularly relevant in cardiovascular patients receiving polytherapy, thus cannot be recomended as a safe and totally without side effects therapeutic option.
In summary, this is a scientifically sound and well-written review. It covers a broad and important topic and is based on a substantial body of recent literature. I recommend its publication after maior revision, particularly the inclusion of comments regarding the translational gap between preclinical and clinical findings, the importance of consumption patterns, and the potential for drug–nutrient interactions.
Author Response
Reviewer 2
This manuscript presents a very interesting and valuable overview of the potential cardioprotective properties of polyphenols. The topic is timely and relevant, both from the perspective of basic research and possible clinical implications.
The authors clearly present the pathophysiological background of cardiovascular diseases, emphasizing the role of chronic inflammation and oxidative stress in the development of atherosclerosis. The classification of polyphenols and the mechanistic discussion are well-structured and comprehensive. Importantly, the manuscript is well-organized, the figures are informative, and the discussion is coherent.
One of the strengths of this review is the use of up-to-date literature. Over 60% of the cited references originate from the last five years, which reflects the current state of knowledge in the field.
Nevertheless, I would like to raise a few points that, while minor, should be addressed to improve the scientific robustness of the manuscript:
1. A large portion of the data presented in the review is derived from preclinical studies using animal models. While this is understandable, it should be explicitly stated that the concentrations of compounds such as resveratrol used in these models often significantly exceed those achievable in humans. Consequently, the translational potential of these findings is limited. This issue has been increasingly emphasized in recent literature, which often describes the gap between promising preclinical results and disappointing clinical outcomes.
Answer: Thanks for this appropriate suggestion; accordingly we have added a new paragraph under the subtitle “caveats in polyphenols” (rows 674-683, in revision mode).
In several sections, the manuscript highlights the beneficial effects of red wine, fruit juices, green tea, and chocolate. However, it should be stressed that their health impact depends not only on polyphenol content, but also on consumption pattern, dose, and dietary context. For example, the culture of drinking red wine with meals, as part of the Mediterranean diet, differs significantly from unhealthy patterns of alcohol consumption. Similarly, frequent intake of fruit juices or sweetened tea may contribute to metabolic disorders due to high fructose content. Therefore, it is essential to emphasize that quantity, context, and overall healthy lifestyle are key factors modulating the effects of polyphenols.
Answer: Thanks again for this right comment; consequentially we have added a second new paragraph under caveats of polyphenols(rows 684-693 in revision mode).
Another important but unaddressed issue is the possibility of interactions between polyphenols and commonly used medications. For example, recent studies have shown that certain polyphenols may interfere with the pharmacokinetics of direct oral anticoagulants (DOACs), potentially altering their efficacy or safety. This should be briefly mentioned, as it is particularly relevant in cardiovascular patients receiving polytherapy, thus cannot be recomended as a safe and totally without side effects therapeutic option.
Answer: We totally agree with the importance of this comment; accordingly we have added a third new paragraph under caveats of polyphenols ( rows 694-710, in revision mode).
In summary, this is a scientifically sound and well-written review. It covers a broad and important topic and is based on a substantial body of recent literature. I recommend its publication after maior revision, particularly the inclusion of comments regarding the translational gap between preclinical and clinical findings, the importance of consumption patterns, and the potential for drug–nutrient interactions.
ANSWER: Thanks for your appreciation.
Reviewer 3 Report
Comments and Suggestions for Authors
This study presents a detailed review of the cellular and molecular mechanisms underlying the beneficial effects of polyphenols in cardiovascular disease (CVD). Its thorough documentation of preclinical and clinical findings integrates existing literature to highlight the therapeutic potential of polyphenols in mitigating oxidative stress, inflammation, and endothelial dysfunction. Furthermore, its discussion of various polyphenol subclasses and their pharmacological activities provides a comprehensive overview of the potential applications in CVD management. However, some points merit further consideration:
1. The study primarily focuses on the purported benefits of polyphenols but does not extensively discuss the potential limitations of their bioavailability and metabolism. Could a deeper analysis of absorption kinetics and metabolic pathways strengthen the assessment of their therapeutic effects and potential toxicities?
2. The reliance on preclinical models raises questions regarding the generalizability of the findings to human populations. Would including additional clinical trials or meta-analyses from existing studies enhance the applicability and robustness of the conclusions drawn?
3. The paper suggests that polyphenols exert their effects via modulation of nitric oxide (NO) production and inhibition of inflammatory pathways. Could further exploration and validation of this hypothesis through mechanistic studies, particularly those involving endothelial cell models, strengthen the conclusions regarding treatment efficacy and resistance mechanisms?
4. While the review mentions dietary sources of polyphenols, it does not sufficiently address the impact of dietary variations on polyphenol effectiveness. Could examining the influence of diet composition, absorption enhancers, or co-administration with other bioactive compounds provide a more nuanced perspective on their clinical utility?
5. The discussion on adverse effects of polyphenols remains relatively brief, with few details on dosage-dependent toxicities. Would a more detailed exploration of dose-response relationships and long-term safety data help refine recommendations for therapeutic applications?
Author Response
Reviewer 3
This study presents a detailed review of the cellular and molecular mechanisms underlying the beneficial effects of polyphenols in cardiovascular disease (CVD). Its thorough documentation of preclinical and clinical findings integrates existing literature to highlight the therapeutic potential of polyphenols in mitigating oxidative stress, inflammation, and endothelial dysfunction. Furthermore, its discussion of various polyphenol subclasses and their pharmacological activities provides a comprehensive overview of the potential applications in CVD management. However, some points merit further consideration:
- The study primarily focuses on the purported benefits of polyphenols but does not extensively discuss the potential limitations of their bioavailability and metabolism. Could a deeper analysis of absorption kinetics and metabolic pathways strengthen the assessment of their therapeutic effects and potential toxicities?
ANSWER. Many thanks for this comment. Yes, a deeper analysis of polyphenol absorption and metabolism is crucial for a more accurate assessment of their therapeutic effects and potential toxicities. While studies often highlight the purported benefits of polyphenols, a more thorough understanding of how they are absorbed, metabolized, and interact with the body can reveal limitations in their bioavailability and identify potential risks associated with their consumption.
So we have added more data (row 102-117 and 193-201, revision mode)
- The reliance on preclinical models raises questions regarding the generalizability of the findings to human populations. Would including additional clinical trials or meta-analyses from existing studies enhance the applicability and robustness of the conclusions drawn?
ANSWER: Thanks for this constructive comment. Yes, including additional clinical trials and meta-analyses would significantly enhance the applicability and robustness of findings related to polyphenols. While preclinical models are valuable for initial investigation, they don't always accurately reflect human physiology and metabolism. Clinical trials and meta-analyses, on the other hand, provide direct evidence of polyphenol effects in human populations, addressing variability and potential interactions. For this purpose we have added a new paragraph at the end of each of the 3 main section (EFFECTS OF POLYPHENOLS ON THE VASCULAR ENDOTHELIUM row 296-299, ANTI-INFLAMMATORY ACTIVITY OF POLYPHENOLS row 348-357, ANTI-THEROGENIC EFFECTS OF POLYPHENOLS row 407-411) dealing with the polyphenols in clinical trials.
- The paper suggests that polyphenols exert their effects via modulation of nitric oxide (NO) production and inhibition of inflammatory pathways. Could further exploration and validation of this hypothesis through mechanistic studies, particularly those involving endothelial cell models, strengthen the conclusions regarding treatment efficacy and resistance mechanisms?
ANSWER: Yes, further exploration and validation of polyphenol efficacy and resistance mechanisms through mechanistic studies, particularly those using endothelial cell models, would significantly strengthen the conclusions. Such studies would provide deeper insights into how polyphenols interact with endothelial cells, the key cells involved in vascular health, and how these interactions affect treatment efficacy and potentially lead to resistance: So we have added a new paragraph at the end of the section EFFECTS OF POLYPHENOLS ON THE VASCULAR ENDOTHELIUM (Row 300-308, revision mode).
- While the review mentions dietary sources of polyphenols, it does not sufficiently address the impact of dietary variations on polyphenol effectiveness. Could examining the influence of diet composition, absorption enhancers, or co-administration with other bioactive compounds provide a more nuanced perspective on their clinical utility?
ANSWER: Yes, focusing on dietary variations, absorption enhancers, and co-administration with other bioactive compounds could provide a more nuanced understanding of polyphenol effectiveness. While a review might mention dietary sources, it often overlooks how the overall diet composition, gut microbiome, and interactions with other compounds influence polyphenol bioavailability and biological activity. Accordingly we have added another paragraph at the end of the section on polyphenols absorption (Row 211-230, in revision mode).
- The discussion on adverse effects of polyphenols remains relatively brief, with few details on dosage-dependent toxicities. Would a more detailed exploration of dose-response relationships and long-term safety data help refine recommendations for therapeutic applications?
ANSWER: Thanks for this very appropriate comment. While polyphenols generally show beneficial effects like antioxidant and anti-inflammatory properties, long-term safety data is still being investigated, particularly for high doses. Some polyphenols can exhibit carcinogenic or genotoxic effects at high concentrations, and interactions with medications or other substances are also possible. Accordingly we have added a new multiarticulate paragraph in the text (rows 605-639, revision mode).
Round 2
Reviewer 2 Report
Comments and Suggestions for Authors
the authors have modified the manuscript, it can now be considered for publication in my opinion